# Novel Therapeutic Approach in PEGylated Chitosan Nanoparticles of Apigenin for the Treatment of Cancer via Oral Nanomedicine

**DOI:** 10.3390/polym14204344

**Published:** 2022-10-15

**Authors:** Md Ali Mujtaba, Nawaf M. Alotaibi, Sultan M. Alshehri, Mohammad Yusuf, Md Khalid Anwer, Mohammad Akhlaquer Rahman, Arshiya Parveen

**Affiliations:** 1Department of Pharmaceutics, Faculty of Pharmacy, Northern Border University, Rafhaa 73213, Saudi Arabia; 2Department of Clinical Pharmacy, Faculty of Pharmacy, Northern Border University, Rafhaa 73213, Saudi Arabia; 3Department of Pharmaceutics, College of Pharmacy, King Saud University, Riyadh 11451, Saudi Arabia; 4Department of Clinical Pharmacy, College of Pharmacy, Taif University, P.O. Box 11099, Taif 21944, Saudi Arabia; 5Department of Pharmaceutics, College of Pharmacy, Prince Sattam Bin Abdulaziz University, Al-kharj 11942, Saudi Arabia; 6Department of Pharmaceutics and Industrial Pharmacy, College of Pharmacy, Taif University, P.O. Box 11099, Taif 21944, Saudi Arabia; 7Department of Pharmaceutics, School of Pharmaceutical Education and Research, Jamia Hamdard, New Delhi 110062, India

**Keywords:** apigenin, Box–Behnken design, antioxidant, chitosan, nanoparticles

## Abstract

The goal of this study was to optimize and formulate apigenin (APG)-loaded pegylated chitosan nanoparticles (PEGylated-CNPs) via ionic gelation techniques using the Box–Behnken design (BBD). Three individual variables, X_1_(chitosan: TPP concentration), X_2_ (PEG-400 concentration), and X_3_ (sonication time), were investigated for their influence on response variables (Y_1_—particle size (PS); Y_2_—drug entrapment efficiency (DEE); and Y_3_—zeta potential (ZP). The optimized formula of APG-PEGylated CNPs was picked from the statistical design and was then examined for physical, morphological, release characterization, anti-oxidant, and anti-tumor potential. The average PS, PDI, %DEE, and ZP were found to be 139.63 ± 5.67 nm, 0.296 ± 0.014, 79.55 ± 3.12%, and 24.68 ± 1.84 mV, respectively. The optimized APG formulation was chosen and reformulated based on the desirability function. Results of the observed and predicted values of responses through the BBD process were found to be nearly identical. The resulting APG-PEGylated CNPs were spherical and smooth, according to surface morphology studies. The release study revealed that PEGylated-CNPs exhibited biphasic release patterns distinguished by an initial burst release of APG only at early phases accompanied by a delayed release near 24 h. Furthermore, APG-PEGylated CNPs demonstrated statistically increased antioxidant activities and cytotoxicity against MCF-7 cells compared to pure APG. Based on the findings, it is possible to conclude that BBD was efficient in optimizing the PEGylated CNPs formulation and recognizing the impacts of formulation variables. In conclusion, the developed formulation has a significant potential for anticancer therapy.

## 1. Introduction

Natural products persist as incredibly valuable sources of new bioactive molecules with pharmacotherapy utility because of the enormous structural complexity found in them. From a long period of time, plants have been used to treat a wide range of diseases such as cancer with natural sources, accounting for more than 60% of today’s anti-cancer medications. Plants, animals, aquatic species, and microorganisms all contribute to drugs, which makes nature an appealing source of new therapeutic entities that could be used as anti-cancer agents [1].

Apigenin (APG), chemically known as 5,7-dihydroxy-2-(4-hydroxyphenyl)-4H-1-benzopyran-4-one, is a flavone found in several vegetables and fruits, which include parsley, celery, onions, oranges, wheat sprouts, corn, rice, tea, and is abundant in chamomile [2]. It has multiple biological effects; in addition, it has potent chemoprotective and chemotherapeutic features for a variety of cancers, including lung, breast, skin, prostate, and colon cancer. The possible mechanism for the anticancer effects of APG could be its capability of inhibiting migration or invasion, endorsement of cell cycle arrest, suppression of proliferation, initiation of apoptosis, or interruption with tumor cellular signal pathways [3,4]. APG has low internal toxicity with no mutagenic properties compared to other plant flavonoids, making it an effective frontrunner for the production of successful cancer therapies [5]. APG is a Biopharmaceutics Classification System (BCS) Class II drug with an aqueous solubility of less than 2.16 g/mL [6]. Unfortunately, its poor water solubility results in poor bioavailability, which significantly limits its clinical use [7]. Therefore, investigating emerging innovations or formulations to improve APG solubility and bioavailability and to enhance antitumor efficacy is, thus, the main issue at present.

Nanoparticles (NPs) are expected to be the key focus in drug-delivery applications; however, NPs are associated with various conjugation properties due to their higher surface area, resulting in in the quickest systemic clearance. To overcome the problem of rapid systemic clearance, NP’s surfaces were coated with a hydrophilic polymer-like chitosan (CS) and polyethylene glycol (PEG). CS is a USFDA-approved and popular polymer because of its low cost of production, biocompatibility, biodegradability, and positive charge. The positive charge of CS enhances interaction as well as penetration inside heterogeneous tumors [8]. CS improves drug penetration by opening epithelial tight junctions and facilitating both paracellular and transcellular drug transport [9]. PEG is also a non-toxic and non-immunogenic polymer that changes a molecule’s hydrophobicity and enhances its aqueous solubility. PEGylation is a well-known method that enhances stability and decreases proteolysis and renal excretion, thereby minimizing dosing frequency and improving pharmacokinetics. Coating with PEG was intended to improve oral nanoparticle bioavailability because it is known to reduce immune system interactions and can eliminate these interactions, thus further increasing systemic half-life [10,11]. One more feature of PEG coating has been shown to reduce nanoparticle agglomeration and improves the therapeutic agent’s safety and efficacy [12,13]. Therefore, the aim of the present study was to create, optimize, and characterize PEGylated chitosan nanoparticles of APG and to evaluate the optimized formulation for its anti-oxidant and anti-tumor potential.

## 2. Methods and Materials

### 2.1. Materials

Apigenin (purity 98%), chitosan medium molecular weight (190–310 kDa and degree of acetylation 85%), and polyethylene glycol (PEG) 400 were purchased from Baoji Guokang Bio-Technology Co., Ltd. Baoji, China. Sodium tripolyphosphate (TPP) and all other reagents and chemicals were of analytical grade in this study.

### 2.2. Preparation of APG-Loaded PEGylated Chitosan Nanoparticles (PEGylated-CNPs)

NPs were prepared by ionotropic gelation and were then homogenized and ultrasonicated [14,15]. For this, CS was dissolved in water that contained 1% *v/v* glacial acetic acid solution. Then, a weighed amount of APG was incorporated to the CS solution, and the added mixture has been homogenized at 12,000 rpm for 10 min. Throughout the process of homogenization, a TPP solution was added drop-wise to the mixture. Based on preliminary research, the CS-to-TPP ratio was determined as 4:1. Instantly, after homogenization, the dispersion was exposed to probe sonication (SONICS Vibra cell VC750, Newton, CT, USA) at 30% amplitude and pulsed at pulse 10 s for 20 min to generate nanoparticles. To keep the temperature increase under control, the dispersion was kept in an ice bath. NPs were coated with PEG 400 immediately after they were formed. Several batches were prepared by using the Box–Behnken design (BBD) to optimize the formulation. The schematic representation for the preparation of PEGylated-CNPs is as follows (Figure 1).

### 2.3. Experimental Design

The BBD is a 3-factor 3-level design experiment that is recommended over others because it requires minimal experimental runs and provides the desired points under the cuboidal space; therefore, the probability of receiving unsuitable results reduced [16,17]. To examine the formulation variables influencing the studied, a three-factor, three-level BBD was applied, in which three formulation variables (amount of CS: TPP, amount of PEG 400, and sonication time) were differentiated at low (−1), middle (0), and high (+1), explored in Table 1. This design necessitates 15 runs with three replicated center points in order to obtain a more uniform estimation of the prediction variance across the overall experimental design. The responses considered in this studied were particle size (PS) (Y_1_), drug entrapment efficiency (DEE) (Y_2_), and zeta potential (ZP) (Y_3_). The BBD was developed by Design-Expert software (Version 12.0, Stat-Ease Inc., MN, USA), which created and analyzed fifteen experimental runs (Table 2). The ranges for each independent factor were preferred from the preliminary study, which can be seen in Table 1.

In order to identify the most desirable mathematical model by using F-tests, Design-Expert produced linear, two-factor interaction (2FI), and quadratic models. Among these, the quadratic model was considered as the best-fitting model with respect to all responses (Table 3). The predicted and adjusted R^2^ should be close with an approximate difference of 0.2 that must be achieved with a “reasonable agreement” [18]. Adequate precision is also one of the significant parameters in predicting the optimum response for a given variable. The chosen model was also subjected to a lack-of-fit test, and the lack of significance observed for this value in comparison to pure errors indicated that the independent variables and their responses had a significant correlation [19]. The polynomial equation obtained by experimental design is expressed mathematically as follows:Y = A _0_+ A_1_ X_1_ + A_2_ X_2_ + A_3_ X_3_ + A_12_ X_1_ X_3_ + A_13_ X_1_ X_3_ + A_23_ X_2_ X_3_ + A_11_ X_1_^2^ + A_22_ X_2_^2^ + A_33_ X_3_^2^(1)
where Y is the dependent responses, A_o_ represents intercept, A_1_ to A_33_ represent the regression coefficients of Y, and X_1_ to X_3_ are presented as independent variables [20]. After producing polynomial equations directly related to dependent and independent variables, the optimization of minimum PS (Y_1_), maximum DEE (Y_2_), and low ZP (Y_3_) was performed using a desirability function to obtain the levels of X_1_, X_2_, and X_3_.

## 3. Physicochemical Characterization of APG-Loaded PEGylated-CNPs

### 3.1. PS, and ZP Analysis

The average PS and ZP of 15 formulations (CNP1–CNP15) were determined with zeta sizer (PSS NICOMP Z3000, Port Richey, FL, USA). Initially, 1 mL of samples was diluted ten times with double distilled water and mixed gently by handshaking to obtain an appropriate scattering intensity; it was measured at a fixed scattering angle of 90°. All measurements were taken in triplicate [21].

### 3.2. Determination of DEE (Drug Entrapment Efficiency) and Loading Capacity (LC)

For determining the DEE, 10 mg of nanoparticles (NPs) was added to 10 mL of acetone to isolate APG into acetone. The samples in acetone were gently ultra-centrifuged at 20,000 rpm for 1 h at 4 °C to collect the supernatant. Then, 10 µL of suspension was diluted to 1 mL with acetone and underwent the process of estimations. The appropriate APG concentrations were obtained with a UV-VIS spectrophotometer (Jenway 6850 double beam spectrophotometer, Cole-Parmer, Saint Neots, UK) [4,22]. The following formula was used to calculate the DEE and LC.
$$\mathrm{DEE}\;(\%)=\frac{\mathrm{Total}\;\mathrm{amount}\;\mathrm{of}\;\mathrm{APG}\;\mathrm{added}\;-\mathrm{Total}\;\mathrm{amount}\;\mathrm{of}\;\mathrm{APG}\;\mathrm{in}\;\mathrm{supernatant}}{\mathrm{Total}\;\mathrm{amount}\;\mathrm{of}\;\mathrm{APG}\;\mathrm{added}}\;\times \;100$$
$$\mathrm{LC}\;(\%)=\frac{\mathrm{Total}\;\mathrm{amount}\;\mathrm{of}\;\mathrm{APG}\;\mathrm{added}\;-\mathrm{Total}\;\mathrm{amount}\;\mathrm{of}\;\mathrm{APG}\;\mathrm{in}\;\mathrm{supernatant}}{\mathrm{weight}\;\mathrm{of}\;\mathrm{NPs}}\;\times \;100$$

### 3.3. Transmission Electron Microscope (TEM) Analysis

The sample for TEM was prepared using phosphotungstic acid staining. The TEM sample was developed by placing a drop of the NPs suspension on a carbon-coated copper grid. The sample was dried and examined with TEM (JEOL JEM1010, Tokyo, Japan).

### 3.4. In Vitro Drug Release

The release study of encapsulated APG from PEGylated-CNPs was studied at two different pH conditions using the dialysis bag diffusion technique, both at 37 °C under sink conditions. The selected dialysis bag (M.W 12 kDa, Sigma–Aldrich, St. Louis, MO, USA) was washed and prepared for study according to the manufacturer’s guidelines. A release study was carried out in PBS at pH 6.8 and pH 7.4 to estimate the drug’s release in tumor surroundings that became slightly acidic and physiological pH, respectively [8,23]. In both studies, samples containing 10 mg APG were installed in dialysis bags and submerged in a beaker of 100 mL PBS with 1% *w/v* Tween 20 retained at 37 °C and stirring at 200 rpm. The sample’s aliquots were withdrawn at predefined time points and examined using a UV-VIS spectrophotometer at 335 nm to estimate the release of APG.

### 3.5. Storage and pH Stability Studies

Physical and chemical stabilities of the optimized formulation were evaluated at two different temperatures 4 ± 2 °C and 25 ± 5 °C for periods of 4 weeks to analyze any changes with respect to PS, PDI, and ZP. The formulation’s stability was also tested at two different pH levels of 6.8 and 7.4.

### 3.6. Determination of Antioxidant Potential of Optimized APG-Loaded PEGylated-CNPs

The antioxidant potential of plant origin materials is defined as their capacity to catalyze the degradation of free radicals, i.e., 1,1-Diphenyl-2-picrylhydrazyl (DPPH), and the ability to significantly reduce their concentration. According to previously published reports, the study was carried out on optimized APG-loaded PEGylated-CNPs and APG suspension to assess their antioxidant potential [24]. The stock solution (10 mg/mL) was diluted with ethanol up until a concentration of 25–250 g/mL. The collected samples (500 L) were added to DPPH solution (0.02 percent in ethanol). The prepared samples were stirred and kept at 25 °C in a dark place for 1 h to carry the complete reaction process. At the end of the reaction, the violet color of DPPH changed to being colorless. The same experiment was repeated with blank PEGylated-CNPs. The samples were spectrophotometrically examined at 517 nm. The percentage of antioxidant activity was calculated as follows.
$$\mathrm{Antioxidant}\;\mathrm{activity}\;(\%\;\mathrm{AA})=\frac{\mathrm{Absorbance}\;\mathrm{of}\;\mathrm{control}-\mathrm{Absorbance}\;\mathrm{of}\;\mathrm{test}}{\mathrm{Absorbance}\;\mathrm{of}\;\mathrm{control}}\;\times \;100$$

### 3.7. Cytotoxicity Study

The impact of various concentrations of pure APG and APG-loaded PEGylated-CNPs on MCF7 cells was analyzed along with MTT (3-(4,5-dimethylthiazol-2-yl)-2,5-diphenyl tetrazolium bromide). The cytotoxic evaluation via the MTT assay is based on the principle of colorimetric tests, which relies on viable cells’ ability to selectively minimize the tetrazolium constituent of MTT to purple-colored formazan crystals [24]. In brief, MCF7 cells were grown into 96-well plates at around 15,000 cells/well density in 100 µL cell media (DMEM, 10% FBS). Seeded cells were kept overnight at 37 ℃ and 5% CO_2_ to enable improved adherence. Various concentrations of pure APG and the formulation were placed in different wells to examine their cytotoxic effects. DMSO was also used to produce stock solutions of the standard and the prepared formulation, which were sequentially diluted in a 96-well plate using serum-free media. The final DMSO concentration was kept below 1% to avoid any unwanted effects on the cells. The same diluted DMSO in the cell culture media was considered as the vehicle control. Following several pilot experiments, a range of concentration was chosen for pure APG and optimized formulations, and the assay was conducted in quadruplicate. After 24 h of exposure to APG, each well, except the blank, was filled with 10 µL MTT solution (5 mg/mL PBS). The treated cells were kept for incubation of 4 h to allow the metabolic process of MTT by viable cells. Cell media was removed from each well, and 100 µL DMSO was incorporated to liquefy the formazan of MTT. After the incubation of 30 min, the plate was read at 570 nm using DMSO as blank. The cellular viability was calculated as the viable cell’s percentage compared with the control group [25]. The dose–effect curve was also used to calculate IC_50_ values, which were then expressed as concentrations (µM).

### 3.8. Statistical Analysis

Data are displayed as Mean ± SD. Raw data were evaluated with Graph Pad InStat demo version (GraphPad Software Inc., La Jolla, CA, USA). Data were examined using one-way ANOVA along with Tukey–Kramer multiple comparison tests to evaluate the statistical significance at different concentration exposures and the control.

## 4. Results and Discussion

In this study, we optimized and developed APG-loaded PEGylated-CNPs for improving therapeutic potential and for investigating the current use of APG in the treatment of cancer. The use of polymeric NPs as a drug carrier was motivated by the fact that it holds unique potential in cancer treatments. This is due to its biocompatibility and safeness with respect to normal tissues, which are attained by hiding the drug’s toxicity. An additional significant benefit of NPs is that they improve drug efficacy at the targeted site because of its permeable properties [8]. Furthermore, CS was chosen as a carrier system for NPs because of its inherent positive charge, as well as other advantages including its biodegradability, biocompatibility, and lower cost. The positively charged surface of NPs targets tumor cells to a greater extent than the negatively charged surface of NPs [26]. They are more readily absorbed by cells, and this may damage cell membranes [27]. The mucoadhesiveness of CS relative to the bio-membrane is due to its adhesive nature. Furthermore, cells preferentially consume positively charged CNPs over negatively or neutrally charged CNPs [28]. The key mechanism is electrostatic interaction between the CS’s NH^3+^ groups and negatively charged DNA’s phosphoryl groups. The positive surface charge allows binding with negatively charged cell membrane and increases permeation.

To confirm the hypothesis, we used a single-step ionotropic gelation method to make CNPs loaded with APG. CNPs were designed by ionic gelation, which requires two aqueous phases mixture, one of which is the polymer CS and another is a polyanion TPP. The ratio of CS to TPP was optimized to be 1:1. The amino group of CS interacts with the TPP to form a nano-sized complex [29]. Furthermore, CNPs were also coated with PEG400 on the surface to enhance their safety and stability. The surface coatings of CNPs with PEG are based on physical cross-linking using electrostatic interactions. The method used here is a simple and gentle method for the preparation of the formulation. It is preferable to use physical crosslinking rather than chemical crosslinking to prevent excipient toxicity and other undesirable side effects. In particular, this method tries to avoid surfactants and organic solvents, resulting in the preparation of a novel formulation that is both safer and more cost-effective.

Preliminary experiments were carried out to ascertain the composition of PEGylated-CNPs. On the basis of preliminary research, three factors were chosen at three different levels: the amount of CS: TPP (X_1_), the amount of PEG 400 (X_2_), and the sonication time (X_3_). According to selected range of the components, BBD resulted in 15 different APG-loaded PEGylated-CNP formulations, and these were examined with selected response variables, i.e., PS (Y_1_), % EE (Y_2_), and ZP (Y_3_), as seen in Table 2.

### 4.1. Designed Experiment Statistical Analysis

The surface response methodology along with BBD is an efficient appliance for the examination of independent variables at a different level with limited experimental runs [20]. In this present study, three-factor three-level factorial designs with respect to responses were investigated. The generated BBD was then used to conduct a series of experiments. The result of measured responses for all formulations are reported in Table 2. The contributing factors and its responses were integrated, with statistical evaluation providing the polynomial equation for evaluating the impact of excipient on responses Y_1_–Y_3_. All dependent variables were analyzed with the quadratic model, and 3D surface plots for responses were created. The quadratic model’s statistical study found significant *p*-values, indicating a better fit for selected responses (Table 4) [30].

### 4.2. Evaluation of Response Surface with Polynomial Equation

The three-dimensional response surface plots shown in Figure 1A–C are helpful for demonstrating the relationship of independent variables with dependent variables. These plots represent the effect of two independent factors on a response at a particular point. Whenever these 3D-plots are critically examined, the characteristic effects of independent variables on each response were observed [16].

### 4.3. Response 1 (Y_1_): Effect of Investigated Independent Variables on PS

The PS of APG-PEGylated CNPs was influenced by the selected independent variables; the average PS of multiple batches of formulations ranged from 120.52 nm to 354.2 nm, as shown in the Table 2. The following polynomial equation describes the mathematical relationship between independent variables and PS for APG-PEGylated CNPs. The coded equation assists in identifying the correlation of factors and PS (Equation (2)).
Y_1_ (PS) = +145.32 + 56.43 A + 18.29 B − 69.42 C + 9.43 AB − 3.00 AC − 8.25 BC + 42.81 A^2^ + 24.35 B^2^ + 46.23 C^2^
(2)

The given equation reflects the quantitative impact of independent variables (A and C) and their interactions in the terms of AB, AC, and BC in the respect of response Y_1_. The coefficient’s *p*-value (0.05) demonstrated that they had a significant effect on Y_1_. The plus sign of the coefficient denotes a synergistic effect, whereas the minus sign denotes the independent variables’ antagonistic effect on response. The factor’s high coefficient value demonstrates that it has a significant impact on the preferred response. All selected responses admirably corresponded with the quadratic model. ANOVA and multiple correlation tests were used to verify the model’s efficiency (R^2^). The quadratic model’s results, along with ANOVA and the multiple correlation test (R^2^), are shown in Table 4.

The F-value 131.32 of this model designated that it is significant. *p*-values less than 0.05 for this model terms are considered as significant. A, B, C, A^2^, B^2^, and C^2^ present as significant model terms, while the model’s terms are not significant if the value is greater than 0.10. The F-value of 3.78 for the lack of fit indicates that it is non-significant in comparison to the pure error. There is an 11.59% probability that a large lack-of-fit F-value is caused by noise. A relatively insignificant lack of fit is satisfactory, and we only want the model to fit. The adjusted R^2^ of 0.9865 is relatively close to the predicted R^2^ of 0.9280; i.e., the difference between them was less than 0.2. The signal-to-noise ratio is calculated by adequate precision. It is better to have a ratio of more than 4. Our signal-to-noise ratio of 37.895 implies an appropriate signal.

Response surface plots were utilized to further clarify the relationship between the dependent and independent variables. Three-dimensional surface response plots were created to better understand the relationship between the variables and PS. The 3D surface plot, illustrated in Figure 1A, showed that increasing the CS:TPP concentration (mass ratio of 1:1) ultimately resulted in the largest PS formation. The relationship between sonication time and PS was found to be negative, with PS decreasing as sonication times increased. The 3D surface plot in Figure 1B demonstrates the influence of CS:TPP concentrations and sonication times on PS, while PEG 400 concentrations and sonication times on the PS of APG-loaded PEGylated CNPs are shown in Figure 1C.

### 4.4. Response 2 (Y_2_): Effect of Independent Variables on % DEE

The outcome of the % DEE study is reported Table 2. It was found that the % DEE ranged between 65.96 and 96.68. The quadratic model was the significant model for the % DEE analysis with a *p*-value of 0.0001, a *p*-value of 0.8763 for the lack of fit, and a % CV of 1.87 after fitting data responses on different models, as shown in Table 3. A polynomial equation (Equation (3)) was used to demonstrate a relationship of different independent factors with % DEE of the APG-loaded PEGylated CNPs.
Y_3_ (% DEE) = +79.13 + 5.62 A +2.67 B − 9.05 C + 0.1400 AB + 1.43 AC + 0.7550 BC +1.87 A^2^ + +1.68 B^2^ +1.75 C^2^
(3)

It can be concluded from equation 3 that CS:TPP (A) and PEG concentrations (B) both had a positive influence on % DEE, whereas the sonication time (C) had an inverse relationship with % DEE. The final model after ANOVA analysis revealed that A, B, and C factors showed significant effects on the % DEE of APG, as all possessed a *p*-value of <0.05. The model’s F-value of 48.35 indicates that it is statistically significant. *p*-values of less than 0.05 are considered significant for model terms. A, B, C, and A^2^ are significant model terms in this study. The model terms are not significant if the value is above 0.10. The lack-of-fit F-value of 0.22 indicates that the lack of fit is not significant in comparison to the pure error. The adjusted R^2^ of 0.9638 is probably close to the predicted R^2^ of 0.9426; i.e., the variation was below 0.2. The signal-to-noise ratio is calculated by adequate precision, and it has a ratio that is more than 4. The ratio of 24.988 indicates that the signal is adequate. The obtained design space is acceptable by using this model.

Interaction patterns are analyzed using a 3D-response surface graph. As shown by the 3D-response surface graph in Figure 2A–C, it is apparent that the percent DEE is significantly affected by CS:TPP concentration, PEG concentration, and sonication time. This could be clarified by the interaction of APG with the hydrophobic group of PEGylated CNPs. Increased CS:TPP and PEG concentrations are assumed to increase the number of hydrophobic groups accessible for interactions with the APG (hydrophobic), resulting in improved drug entrapments in NPs [31].

### 4.5. Response 3 (Y_3_): Effect of Investigated Independent Factors on ZP

The surface charge of the APG-loaded PEGylated CNPs formulations ranged from +19.41 to +26.56 mV (Table 2). The positive charge on the NP’s surface indicates the existence of freely ionized amino groups that is necessary for electrostatic repulsion between particles to produce stable nano-dispersions. The measured response, Y_3_, has a mathematical concept of a polynomial equation, which is enumerated below.
Y_3_ (ZP) = +23.88 + 2.56 A + 1.02 B + 0.0762 C − 0.0375 AB + 0.0450 AC + 0.1475 BC − 1.42 A^2^ + 0.6650 B^2^ −0.2275 C^2^(4)

The obtained mathematical equation can be used to determine the relative effects of these factors by equating the factor’s coefficients. The concentration of CS:TPP and PEG had more significant effects on ZP. The model’s F-value of 116.01 suggests that it is really significant. Model terms with *p*-values below 0.05 are significant. This case show that that A, B, A^2^, and B^2^ are considered as significant model terms. If the value is higher than 0.10, the model’s terms really are not significant. The F-value for the lack of fit is 1.48, which indicates that the lack of fit is insignificant in comparison to the pure error. An insignificant lack of fit is acceptable, and we desired to fit the model. The adjusted R^2^ of 0.9848 is probably close to the predicted R^2^ of 0.9390; the difference is below 0.2. Adequate precision was discovered at 35.779, indicating that the obtained model is used to processed the design space. 3D surface plots that analyze the impact of independent variables on ZP are reported in Figure 3A–C.

### 4.6. Formulation Optimization for the APG-PEGylated CNPs

The impact of independent variables on responses, as well as the levels of such factors, was decided using the RSM statistical optimization process. Response optimization was used to accomplish an improved formulation with adequate PS, ZP, and high DEE. The selected optimized formulation produce values of independent variables, viz., at X_1_ (CS: TPP concentration) of 0.5% *w/v*, X_2_ (PEG 400 concentration) of 0.75% *w/v*, and X_3_ (sonication time) of 20 min, which granted predicted values for selected responses. Three formulation batches of optimal content were prepared, and their responses for each formulation were examined to confirm the model’s appropriateness for prediction. The optimized formulation components, as well as the experimental and predicted values including all dependent responses, are shown in Table 5. The experimental values were found to be very close to the predicted values, denoting the achievement of the BBD coupled with a preferable function for formulation evaluations and optimizations. For predicted versus observed values, the R^2^ values (>0.9) showed a linear correlation across all dependent variables, denoting the superior model’s reliability (Figure 4).

### 4.7. Characterization of Optimized APG-PEGylated CNPs

The particle size distribution of NPs play a characteristic features in biodistribution, and as a drug carrier, its required a small particle size (i.e., a low PDI value) [4]. The PS distribution spectrum for the optimized APG-PEGylated CNPs is presented in Figure 5A, expressing the average PS of 139.63 ± 5.67 nm and PDI of 0.296 ± 0.014. The EE% and LC% of the formulation were found at 79.55 ± 3.12% and 37.78 ± 2.46%, respectively. The surface morphology of the optimized nanoparticle was examined by TEM (Figure 5B). The NPs were found to be nearly spheroidal vesicles in shape, with a size <200 nm, which matches the results of Zeta Sizer’s PS analysis. The image also revealed that the NPs maintained their shape and that the vesicle’s integrity is preserved. The layer seen around the nanoparticle’s core could be the PEG molecules tied to the CS polymer.

### 4.8. In Vitro Dissolution Studies

To estimate the in vivo release profile of APG, in vitro dissolution analyses were conducted in PBS at pH 6.8 and 7.4. In comparison to free drug suspensions, PEGylated CNPs showed controlled drug released, as observed in Figure 6A,B. A biphasic release was also found with an initial burst release of APG in the early phase, followed by a delayed release of APG in the later phase (up to 24 h). The optimized CNPs showed a sustained release, but the rate was a little higher in pH 6.8 than in PBS pH 7.4. In 8 h of study, about 74% of the drug was released at pH 6.8 and about 62% was released at pH 7.4. The burst effect could have been caused by APG elution near the surface during nanoparticle preparation, which then quickly dispersed when nanoparticles interacted with the dissolution medium. Because of the swelling or deterioration of the polymer, APG was slowly released later on. The remaining APG in nanoparticles did not entirely release until particles have been completely eroded or dissolved in the dissolution media because of the interaction between both the residual APG and the few amine groups of chitosan.

### 4.9. Storage and pH-Dependent Stability Studies

Colloidal systems require a high level of stability. Physical instability is frequently a big obstacle to their clinical application. Optimized PEGylated CNPs were stored for three months at temperatures of 5 and 25 °C to assure physical stability. The estimation of stability was carried out based on PS, PDI, and ZP (Table 6). The measured features of prepared formulation were stable and had a statistically insignificant difference, indicating that CNP formulations remained stable during these time periods.

When the formulation comes in direct contact with different pH conditions, it may experience physical changes that cause particle aggregation. As a result, the formulation’s physical stability was tested at various pH levels to which it could come into contact. The stability characteristics of CNPs in various experiments is shown in Table 6. The PS, PDI, and zeta potential were all insignificantly different, ensuring the stability of stored formulations under changing pH conditions [32].

### 4.10. In Vitro Antioxidant Activity: DPPH Assay

The biological activity of APG is influenced by antioxidant activities. The DPPH assay was applied to measure the antioxidant activity of the APG-PEGylated CNPs formulations in vitro, which relies on the lowering of the DPPH radical in the existence of an antioxidant molecule. DPPH is an incredibly sensitive and commonly used antioxidant assay that is used to evaluate antioxidant activities of a wide range of drugs and drug-delivery systems [16,33]. The test results are reported in Table 7, and it is undeniable that APG has anti-oxidant properties. Because of the presence of CS, weak antioxidant activities were also identified in the blank formulations (13.8 ± 3.1%). The CS alone showed some scavenging activities in a previous study by Anraku M. et al. [33]. These findings also show that CS has antioxidant properties and point to potential pharmaceutical applications. These findings indicate that the APG-PEGylated-CNPs formulation had higher radical-scavenging actions than the pure APG suspension. Overall, these findings show that APG-loaded PEGyalted-CNPs are more effective at scavenging oxygen- and nitrogen-centered radicals and have higher antioxidant capacities.

### 4.11. Cytotoxicity Study

The possible features of the cytotoxic activity study in MCF-7 cells are depicted in Figure 7. When compared to pure APG, PEGylated CNPs composite exhibited significantly greater concentrations and time-dependent cytotoxicity at 24 h of study. Data reveal that the lower concentrations of both APG standard and APG-PEGylated CNPs formulation have MCF7 cell viability, which decreases as the concentration increases (Figure 7). When considering the effects, it seems that the lower concentrations of the APG formulation had improved growth inhibitory effects in MCF7 cells than the standard APG. These influences and comparisons are depicted in Figure 7. After 24 h of treatment against MCF-7 cells, the IC_50_ values of PEGylated CNPs and pure APG were observed to be 162 ± 14.54 µM and 1834.1 ± 55.74 µM, respectively. When compared to standard APG, data from the cell viability assay revealed that the PEGylated-CNPs formulation containing APG significantly enhanced these impacts in terms of lowering the IC_50_. The APG inhibits the growth of MCF7 cells in a concentration-dependent manner, as reported by cell viability assays or MTT assays. However, that could be time-dependent, and the current study did not require a time-dependent cell viability assay.

This can be explained by the fact that positively charged surface NPs have a higher chance of attacking tumor cells and are more easily gained by cells, potentially causing cell membrane destruction [29,30]. The important mechanism is the electrostatic interaction of the positively charged NH^3+^ groups of CS with the negative charge entity of DNA. They seem to have a total positive surface with respect to NPs, which allows them to bind with plasma membrane and increases the permeation through it [8].

## 5. Conclusions

In the present inspection, BBD was successfully used to design and formulate PEGylated APG-encapsulated CNPs. BBD had been used to generate design spaces, and three independent factors were used to achieve an optimized formulation that resulted in a smaller size, optimal ZP, and a higher % DEE. The single-step ionotropic gelation technique was used to make CNPs. PEG 400 was also applied to the surface of the NPs to ensure their safety and stability. For APG-PEGylated CNPs, the optimized formula yielded PS, ZP, and % DEE of 139.63 ± 5.67 nm, 24.68 ± 1.84, and 79.55 ± 3.12 respectively. TEM micrographs revealed a smooth surface and a spherical shape. According to the release study, PEGylated CNPs had a sustained release pattern for 24 h of periods. Once compared to pure APG, APG-loaded PEGylated CNPs had significantly higher antioxidant capacity. The formulation showed increased dose-dependent antitumor activities in breast cancer cells in comparison to pure APG. As a result, the study’s conclusive result is that PEGylated CNPs improved the therapeutic potential and expanded the current use of novel formulations of APG in cancer treatments.

## Data Availability

Data are contained within the article.

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
