# Peer review of "Novel Therapeutic Approach in PEGylated Chitosan Nanoparticles of Apigenin for the Treatment of Cancer via Oral Nanomedicine"

_polymers, 2022, doi:10.3390/polym14204344_

Round 1

Reviewer 1 Report

The work by Mujtaba et al. presents how chitosan nanoparticles can be exploited to deliver a pharmacological compound of natural origin. Prior to publication, the authors need to address the following concerns:

1. The authors failed to present the purity of used materials.

2. Details on the chemistry for the synthesis should be shown schematically to guide the readers.

3. The characterization of as-synthesized nanoparticles is insufficient. The authors need to provide additional microscopic, thermal, and spectroscopic data, otherwise is really hard to believe that the synthesis and encapsulation were successful. 

4. The authors claimed that they conducted apoptotic and antitumor assays, which is incorrect. They only performed a cytotoxicity assay. For apoptosis, they need to conduct analyses with biological makers such as Annexin V and for antitumoral activities, they need to test the therapy with spheroids.

5. In Figure 7, it is not clear the difference between the standard and the formulation concentrations. Please clarify.

Reviewer 2 Report

This manuscript reports how to optimize and formulate apigenin loaded pegylated chitosan nanoparticles, and investigate 1-chitosan: TPP concentration, 2-PEG-400 concentration, and 3-sonication time, their influence on response variable Y1- particle size, Y2- drug entrapment efficiency, and Y3- zeta potential. The physical, morphological, release characterization, anti-oxidant, and anti-tumor potential were further studied. The results are very good and interesting. However, some points of the manuscript should be improved. Specific comments are given below.

1.    The molecular weight of chitosan should be offered.

2.    Drug loading content should be measured.

3.    CNPs were also coated with PEG400 on the surface to enhance their safety and stability. The mechanism of formation of APG-loaded PEGylated-CNPs should be further discussed.

4.    The statics of TEM micrograph of optimized APG-PEGylated CNPs formulation should be offered.

5.    At lines 67, it is recommended to indicate the type of chitosan, ordinary chitosan is insoluble in water and cannot be called a hydrophilic polymer.

6.    It is recommended to add data such as NMR magnetism or characterize the synthesis results.

7.    The IC50 values of PEGylated CNPs and pure APG were observed to be 162 454 14.54 M and 1834.1 55.74 M, respectively. The results are wrong.

8.    Please carefully check the text for writing and grammar.

Reviewer 3 Report

Recommendation: Minor revisions needed. 

Comments: 

The paper by Mujtaba et al. contributes to an therapeutic approach to formulate APG-loaded pegylated chitosan nanoparticles (PEGylated-CNPs) via ionic gelation techniques. The article gives an interesting scientific perspective on assessing Three individual variables' impact on these compound. Some issues should be addressed prior to publication. 

  1. Abstract. Some font sizes are not correct, please change them accordingly. 

  1. Page 2, line 94. Some font sizes are not correct, please change them accordingly.

  1. Table 1 and 3. The content in this table is not showing correctly, please revise.

  1.  
  1.  
